# Fungal Pathogens Affecting the Production and Quality of Medical Cannabis in Israel

**DOI:** 10.3390/plants9070882

**Published:** 2020-07-13

**Authors:** Shachar Jerushalmi, Marcel Maymon, Aviv Dombrovsky, Stanley Freeman

**Affiliations:** 1Department of Plant Pathology and Weed Research, ARO, The Volcani Center, Rishon LeZion 7505101, IL, Israel; shacharje1@gmail.com (S.J.); marcelma@volcani.agri.gov.il (M.M.); aviv@volcani.agri.gov.il (A.D.); 2The Robert H. Smith Faculty of Agriculture, Food and Environment, The Hebrew University of Jerusalem, Rehovot 7610001, Israel

**Keywords:** *Alternaria alternata*, *Botrytis cinerea*, crown root, *Fusarium oxysporum*, medical cannabis, molds, pathogens, plant disease, stem wilt

## Abstract

The use of and research on medical cannabis (MC) is becoming more common, yet there are still many challenges regarding plant diseases of this crop. For example, there is a lack of formal and professional knowledge regarding fungi that infect MC plants, and practical and effective methods for managing the casual agents of disease are limited. The purpose of this study was to identify foliar, stem, and soilborne pathogens affecting MC under commercial cultivation in Israel. The predominant major foliage pathogens were identified as *Alternaria alternata* and *Botrytis cinerea*, while the common stem and soilborne pathogens were identified as *Fusarium oxysporum* and *F. solani*. Other important fungi that were isolated from foliage were those producing various mycotoxins that can directly harm patients, such as *Aspergillus* spp. and *Penicillium* spp. The sampling and characterization of potential pathogenic fungi were conducted from infected MC plant parts that exhibited various disease symptoms. Koch postulates were conducted by inoculating healthy MC tissues and intact plants with fungi isolated from infected commercially cultivated symptomatic plants. In this study, we report on the major and most common plant pathogens of MC found in Israel, and determine the seasonal outbreak of each fungus.

## 1. Introduction

Cannabis (*Cannabis sativa* L.) is an annual herb from the Cannabaceae family, in which cannabis is the only genus. The common understanding today is that one species exists, containing two subspecies: *C. sativa* subsp. sativa and *C. sativa* subsp. indica. However, nowadays, all cannabis cultivars are hybrids of the two, and it is no longer possible to distinguish between the species [1]. 

In recent years, the popularity and use of medical cannabis (MC) has expanded rapidly throughout Israel and worldwide [2]. As the demand for MC increases, many more farms have been established, and with the growing cultivation intensity, challenges and problems have arisen. These challenges are varied and can be elaborated on as follows: (i) since MC is designed for medical purposes, there are strict regulations regarding the growth, quality, and general standards pertaining to the final product, such as pesticide residues, limits on total yeast and mold (TYM) colony forming units (CFUs), and mycotoxin levels present in MC dried inflorescences [3]; (ii) a lack of theoretical knowledge of MC plant pathogens and disease reduction methods exists, as there is a lack of scientific research on this subject, and most of the information available refers to the fiber type plants (hemp) grown outdoors, since drug type plants were illegal in the past [4]; (iii) years of MC inter-crossbreeding and the use of cuttings in commercial farms have led to low plant genetic diversity and increased susceptibility to plant pathogens and pests [5]. Specifically, it has been reported that drug type MC plants tend to be less resistant when grown in high concentrations compared to fiber type plants [6]. This should be seriously considered, since intensive MC cultivation refers to the drug type cultivars that contain many plant therapeutic phytocannabinoid compounds [7,8].

To date, there are over 88 known fungal species affecting MC and hemp plants at all growth stages, not including storage pathogens [1]. Of these fungal pathogens, the most common inflorescence disease is gray mold, caused by *Botrytis cinerea*, which thrives under high humidity and cool to moderate temperatures, and peaks in maritime conditions [1]. Based on reports from MC farms and from the Israeli Agriculture Ministry reports, crop losses caused by plant diseases are estimated at over 10%. In recent research examining fungal pathogens isolated from drug type cannabis inflorescences in Canada, six main pathogens were recovered: *B. cinerea*, *Fusarium solani*, *F. oxysporum*, *F. equiseti*, *Penicillium copticola*, and *P. olsonii* [9]. In a similar study examining potential soilborne and crown rot cannabis pathogens, four different fungi were reported to cause disease symptoms in MC plants via root inoculations: *F. oxysporum*, *F. brachygibbosum*, *F. solani*, and *Pythium aphanidermatum* [10]. Nevertheless, there is a lack of scientific research and knowledge regarding MC plant diseases, especially in areas where different climatic and ecological niches exist.

In addition to plant physiological damage and crop losses, some MC pathogens impose a direct health hazard to patients that consume infected inflorescences. For example, *B. cinerea*, which is one of the most common pathogens of MC, is a known allergen and can lead to harsh reactions in humans [11]. Furthermore, air samples from the lungs of workers from MC farms have contained a significantly higher than normal concentration of microorganisms, especially different kinds of Ascomycota, of which *B. cinerea* was the most common [12]. This suggests that exposure to fungal spores via MC consumption may lead to potential disease complications. Another more extreme example is a case of two immunocompromised patients that inhaled MC as an analgesic and developed invasive pulmonary aspergillosis [13]. In retrospect, it was found that the MC supplied to these patients was contaminated with spores of *Aspergillus* spp., which was likely the source of infection [13]. These examples highlight the importance of cannabis plant disease management protocols, including the procurement of healthy and safe products for MC patients. 

This research was conducted to obtain a better understanding of the most common diseases of MC cultivated in Israel. The major pathogens affecting roots, stems, leaves, and flowers from commercial farms throughout Israel were identified and characterized by morphological and molecular methods, in order to prepare suitable disease management protocols.

## 2. Results

### 2.1. Fungal Isolations and Characterization

In order to determine which fungi pose an agriculture threat to MC cultivation in Israel, infected MC plants were collected, and fungal cultures were isolated from five different commercial farms (Table 1). One hundred and one fungal cultures were identified and characterized by morphological and molecular criteria, with percentages of the isolates as follows: *Alternaria alternata* (18%)*, Fusarium oxysporum* (8%), *Fusarium solani* (6%)*,* other *Fusarium* spp. (11%), *Botrytis cinerea* (5%)*,* and *Trichoderma* spp. (6%). Additional fungi were identified in varying but lesser frequencies (Figure 1; Table 2 and Appendix A). Other fungi of interest that were identified belonged to *Aspergillus* spp., *Penicillium* spp., and *Trichothecium roseum*. Sampled fungal populations changed throughout the year, according to the seasons in which they were collected (Figure 1). In the spring season (March–May), 24% of the identified isolates belonged to *B. cinerea,* 18% belonged to *A. alternata,* and 18% to *F. oxysporum.* In the summer season (June–August), 22% of the sampled isolates belonged to *F. oxysporum*, and 17% to *F. solani*, while only 17% were identified as *A. alternata.* During the autumn season (September–November), a similar pattern to the general results was found. In the winter season (December-February), only 9% of the fungal population belonged to *A. alternata,* while during this period no *Fusarium* spp. were isolated. Symptoms in the affected plants of each specific fungal disease were similar during the different seasons, but varied in their severity levels (Figure 1). 

### 2.2. Koch Postulate Assays on Detached Tissues 

In detached inflorescences, five of the tested fungal species (*Aspergillus fumigatus*, *A. niger*, *Cladosporium halotolerans*, *C. sphaerospermum*, and *Penicillium olsonii*), out of 19 total, did not exhibit disease symptoms (Table 1). The other 14 fungi caused varying symptoms in infected inflorescences. *Alternaria alternata* caused white-colored lesions in inflorescences and sugar leaves (leaves that are part of the inflorescences) (Figure 2A). *Botrytis cinerea* caused similar symptoms to those of *A. alternata*, including a light gray-colored mold covering the inflorescences, without lesion development (Figure 2B). *Aspergillus flavus* symptoms appeared as a light gray mold with visible yellow conidia (Figure 2C). Symptoms caused by *A. westerdijkiae* were similar to those of *A. flavus*, although mold development was more prominent, also covering the sugar leaves; conidia possessed a yellow–pink coloring (Figure 2D). Both *Cladosporium* spp. and *C. tenuissimum* caused a light white–gray-colored mold that developed on inflorescences (Figure 3E,F). *Penicillium citrinum* caused a light white-colored mold covering whole inflorescences and sugar leaves (Figure 2G). *P. steckii* symptoms were visible mainly as the development of black conidia throughout inflorescences (Figure 2H). *Trichothecium roseum* developed an initially white-colored mold, followed by the appearance of pink conidia (Figure 2I). *Fusarium equiseti* symptoms were characterized by the massive development of white-colored mold, completely covering inflorescences and sugar leaves (Figure 2J). Control, sterile potato dextrose agar supplemented with chloramphenicol (PDAC) plug-inoculated leaves and inflorescences remained healthy (Figure 2K). Detached leaves were inoculated with the most common pathogens (Figure 3). In leaves inoculated with an agar plug of *A. alternaria* (Figure 3A) and *B. cinerea* (Figure 3B), typical brown-colored lesions were observed that spread very rapidly over the leaf surfaces. Control leaves inoculated with sterile PDAC plugs remained healthy (Figure 3C).

### 2.3. Koch Postulate Assays on Intact Plants

Koch postulate assays were performed on intact plants using both *Alternaria alternata* and *Botrytis cinerea*, since they were the two most frequently isolated inflorescence pathogens (Figure 4). Both fungi (Figure 4A: *A. alternata*, and Figure 4B: *B. cinerea*) produced very similar symptoms of brown–gray-colored molds growing on the inflorescences.

### 2.4. Koch Postulate Assays of Soilborne Pathogens on Seedlings 

*Fusarium oxysporum, F. proliferatum*, and *F. solani* were tested for pathogenicity on MC seedlings (Figure 5). All three fungi produced wilt symptoms on seedlings after approximately 4 weeks following inoculation. Symptoms first appeared as a loss of turgor and curling of the leaves, without loss of plant color, followed by drying, wilting, and death of the seedling (Figure 5). Wilt symptoms and plant mortality were observed in approximately 66% and 80% of the seedlings inoculated with *F. proliferatum* and both *F. oxysporum* and *F. solani*, respectively (Figure 5B,D). Reduced and stunted growth was observed in the remaining plants, while the control, non-inoculated, saline-watered seedlings remained healthy (Figure 5A,C). 

In all Koch postulate assays, the corresponding fungi were re-isolated from infected plant parts and verified as the casual agents of disease by morphological characteristics, except for fungi that did not cause disease symptoms (Table 1).

## 3. Discussion

The use of medical cannabis (MC) has increased immensely over the past decade [2]. One of the main reasons for the increased interest in MC are the therapeutic qualities of this plant [7,8,14,15]. Diseases affecting MC have not been studied thoroughly to date, mainly due to restrictions placed on MC cultivation and limited research results [4,9]. With the growing popularity of MC use and expansion of intensive cultivation of cannabis, the presence of MC pathogens is expected to increase accordingly [4], highlighting the importance of identification in order to manage the associated diseases. The main goal of this work was to characterize and identify the threats and difficulties exposed by intensive MC cultivation, from a fungal plant disease perspective, in order to expand our knowledge regarding these concerns under commercial cultivation in Israel. 

In order to achieve these goals, it was necessary to first detect the casual agents of disease in MC plants. Therefore, we collected fungal isolates from infected plants from various commercial MC farms throughout Israel, and tested their pathogenicity using Koch postulate assays on both detached tissues and intact plants (Table 1). *Alternaria alternata* was found to be the most common fungus isolated in general, most frequently comprising the fungal isolates sampled from infected MC inflorescences and leaves (22%), followed by *Botrytis cinerea* (7%), and *Fusarium oxysporum* (5%) (Figure 1; Appendix A). 

These results are interesting compared to those found in a similar research conducted in Canada, in which the most common MC inflorescence pathogens found were *B. cinerea, Fusarium solani, F. oxysporum, F. equiseti*, *Penicillium olsonii*, and *P. copticola*, whereas *A. alternata* was not detected [9]. This difference could be explained in part by the variation in location, environment, and climate, highlighting the importance of local research and data collection. Indeed, most *Alternaria* spp. are reported to grow ideally at relatively elevated temperatures, as high as 27 ºC [1]. 

Some *Alternaria* species are phytopathogenic, secreting specific, host-selective toxins referred to as different pathotypes, especially the *A. alternata* species that are specific pathogens of certain hosts [16,17,18,19]. These pathotypes cause a variety of severe diseases in economically important crops, such as tomato, tangerine, strawberry, Japanese pear, apple, tobacco, etc., that can lead to extensive yield losses. For example, in August 2000, an outbreak of leaf spot disease of broadleaf tobacco (*Nicotiana tabacum* L.) caused by *A. alternata* led to a loss of 75% and 89% of the total plantation areas in Connecticut and Massachusetts, respectively [20]. Not only that, but exposure to *A. alternata* has been associated with asthma, allergic rhinosinusitis, and skin infections in humans, and some of its mycotoxins are mutagenic, making the detection and elimination of this fungus from MC even more crucial [21,22,23]. 

*Botrytis cinerea* on the other hand, while also necrotrophic, has a very broad host range, reportedly attacking a total of 596 genera of vascular plants, representing over 1400 plant species [24]. In strawberries, gray mold caused by *B. cinerea* is the most common reason for fruit rejection, leading to extensive economical losses [25]. *B. cinerea* outbreaks have led to serious losses to chickpea growers—e.g., in Argentina, a gray mold outbreak led to a loss of 95% of total yield, while in Bangladesh, gray mold outbreaks in 1988 and 1989 led to losses of 80% and 90% of total yields, respectively [26]. One of the main difficulties in managing diseases caused by this fungus is that many classes of fungicides fail to control pathogen outbreaks, due to their genetic plasticity and resistance mechanisms [27]. Furthermore, *B. cinerea*, which has been shown from this study to be a common pathogen of MC in Israel, is a known allergen, and can lead to detrimental effects in humans [11].

A pattern of seasonality in the appearance of different fungi affecting MC was found in our work, indicating that *A. alternata* was most common during the spring and autumn, in which temperatures usually are not as low as in winter, and elevated humidity levels prevail (Figure 2). Understanding the seasonal appearance of fungal pathogens may contribute significantly in managing plant diseases in MC via agrotechnical means, i.e., planning the growth cycle in such a way that all inflorescences are harvested before optimal conditions for the appearance of the major pathogens, thus reducing yield loss and preventing extensive infections. 

In a recent study conducted in Canada, the two most common soilborne fungal pathogens affecting MC isolated from roots and stems were identified as *Pythium aphanidermatum* and *Fusarium* spp. [10]. While *Pythium* spp. pathogens were not detected in our work, the most frequent fungal pathogens that were isolated from MC stems and roots in our study were *F. solani* (21%), *F. oxysporum* (11%), and *Lasiodiplodia theobromae* (11%) (Figure 1; Appendix A). This further highlights the difference in climatic and agrotechnical conditions that may have an important effect on the appearance and presence of certain pathogens at different locations worldwide. *F. oxysporum* is a very common plant pathogen causing vascular wilt in many different plants, containing over 120 forma species, each with its own specialized hosts [28]. One of the most recent examples of *F. oxysporum* concerns is that of banana Fusarium wilt, caused by *F. o*. f. sp. *cubense*, which is considered one of the most destructive diseases of this crop in history [29]. *Fusarium* spp. are also known to generate mycotoxins, such as fumonisins, which can harm mammals consuming infected cereals [30,31,32]. Consumption of infected cereals has been associated with esophageal cancer [30]; leukoencephalomalacia in mammals; and hepatotoxic, carcinogenic, and apoptosic (programmed cell death) effects in the liver of rats [32,33]. 

Another important group of trichothecene mycotoxins that are produced by certain fungi were found to be pathogenic to MC (Table 1). These important mycotoxins produced by *Fusarium* and *Trichothecium* spp. are deoxynivalenol (DON), nivalenol (NIV), and T2 toxin, known for their ability to inhibit eukaryotic protein synthesis. DON ingested in high doses caused nausea, vomiting, and diarrhea in animals [33]. The T2 toxin is associated with alimentary toxic aleukia disease, and is symptomatic with inflammation of the skin, vomiting, and hepatic tissue damage [33]. Thus, the potential presence of mycotoxins in MC material sets an example for how crucial it is to deliver a clean and safe product to patients, as consuming infected products may have serious and adverse consequences especially in immunocompromised individuals. 

The accumulation of information regarding common plant pathogens in MC may lead to the development of novel strategies and agrotechnical approaches regarding disease management, and may assist with more precise and helpful regulations regarding phytosanitary protocols for MC products. In Israel, there are strict limitations to the use of pesticides in MC. The use of biocontrol agents is also limited, since CFU levels are likely to increase in treated MC products beyond those permitted by regulators [9]. Therefore, research in regards may provide effective measures for disease control considering the current restrictions. 

The results of this research have helped promote these goals and provided the necessary information regarding specific pathogens of MC in Israel. For example, *Alternaria alternata* was identified as a major inflorescence pathogen in Israel that not only damages MC yields, but also produces toxins, such as alternariol, alternariol monomethyl ether, and other altertoxins, that have deleterious effects on humans (Jerushalmi et al., data not shown) [34]. Identifying the common pathogens that affect MC plants, which plant tissues they infect, and during which seasons they appear, is crucial for MC disease management, leading to increased yield, better quality, reduced application of fungicides, and a safer healthier product. 

## 4. Materials and Methods

### 4.1. Isolation of Fungi and Growth Media

Infected cannabis plant material was collected from five commercial MC farms from different locations in Israel between the years 2017–2019, and 101 potential pathogenic fungi were isolated from different symptomatic tissues, including inflorescences, leaves, stems, and roots, in the laboratory. Samples were numbered, plant material was documented, and details of samples were recorded (Appendix A). The isolation of fungi from the infected plants was performed by sectioning 1 × 1 cm pieces of relevant infected tissues and surface sterilizing by immersing plant material in 70% ethanol for 30 s, followed by 1% sodium hypochlorite (NaClO) for 3 min. Thereafter, plant tissues were dried on sterile paper towels, placed aseptically on Petri plates containing PDAC 1.2% medium (potato dextrose broth (PDB): 24 g/L; agar (Difco, Franklin Lakes, NJ, USA): 12 g/L; chloramphenicol (ACROS Organics, Geel, Belgium): 0.25 g/L), and kept at room temperature to allow fungi to develop. After 3–5 days, fungal mycelia growing from affected plant material were transferred to new Petri plates containing PDAC 1.5% medium, and allowed to develop for 3–5 days. All subsequent fungal cultures were single-spored, according to Choi et al. [35], before identification and characterization by morphological and molecular methods. 

### 4.2. Fungal Identification and Characterization

The isolated fungi were morphologically identified and characterized using light microscopy, and further verified using molecular methods by polymerase chain reaction (PCR) amplification. For molecular analyses, DNA was extracted from all the putative pathogens, as follows. Fungal isolates were re-cultured in liquid FMM medium at 25 °C for a week, in order to harvest dried mycelium. FMM medium contained D-Glucose (Dextrose) (Duchefa Biochemie, Haarlem, The Netherlands), 10 g/L; 20 × salt solution, 50 mL/L; trace elements, 1 mL/L; 20 × salt solution containing NaNO_3_ (120 g), KCL (10.4 g), MgSO_4_•7H_2_O (10.4 g); and KH_2_PO_4_ (30.4 g per 1 L of ddH_2_O). The trace element solution contained ZnSO4•7H2O, 27.5 g; H_3_BO_3_, 13.75 g; MnCl_2_•4H_2_O, 6.25 g; FeSO_4_•7H_2_O, 6.25 g; CoCl_2_•5H_2_O, 2 g; CuSO_4_•5H_2_O, 2 g; (NH_4_)6MO_7_O2_4_•4H_2_O, 1.375 g; and Na_4_EDTA, 62.5 g per 1 L of ddH_2_O. The liquid medium was filtered through 54 mm filter paper and freshly harvested mycelium (100–200 mg), then placed in 2 mL tubes. Quantities of 500 µL phenol/chloroform/isoamyl alcohol (1:24:25) solution, two sterile metal beads (SPEX Sample Prep, Metuchen, NJ, USA) 3 mm diameter, and 700 µl of breaking buffer (comprised of 20 mL Trition × 100, 100 mL SDS, 20 mL NaCl, 10 mL Tris–Cl (pH 8.0), and 2 mL EDTA) and 848 mL of ddH_2_O were added to each tube. Tubes were sealed and placed in a Geno/Grinder 2010 (SPEX Sample Prep, NJ, USA) and centrifuged for 90 s at 1600 revolutions per min (RPM). The contents were further centrifuged (BIOFUGE Fresco, Heraeus, Hanau, Germany) at 4 °C for 10 min at 1300 RPM, and 500 µL of the supernatant was transferred to new tubes. Quantities of 50 µL of 3M Na acetate pH 5.2 and 1 mL cold 100% ethanol were added to each tube. DNA was precipitated by centrifugation at 13,000 RPM for 10 min, and washed twice with 70% ethanol. Pellets were dried in a laminar-flow chamber, resuspended in 50–100 µL TE buffer (1 mM EDTA, 10 mM Tris) [36]. DNA was concentrated to 100 ng/μL using a Nanodrop spectrophotometer ND-1000 (Thermo Scientific, Waltham, MA, USA) before molecular identification. PCR was performed on all fungal DNA samples (Appendix A) using the internal transcribed spacer (ITS) 4 (5′-TCCTCCGCTTATTGATATGC) and ITS 5 (5′-GGAAGTAAAAGTCGTAACAAGG) primers, as previously described [36]. In order to determine and further verify the species level of *Fusarium* isolates, additional primers were used [37]. The reference primers in this case were elongation factor (EF) 1-728F (5′-CATCGAGAAGTTCGAGAAGG) and EF1-986R (5′-TACTTGAAGGAACCCTTACC) with the same reagent composition and PCR amplification conditions as that for ITS 4–5. For further identification of the *Fusarium oxysporum* complex, PCR amplification was conducted using primers of the nuclear ribosomal intergenic spacer region (IGS), iNL11 (5′-AGGCTTCGGCTTAGCGTCTTAG), and iCNS1 (5′-TTTCGCAGTGAGGTCGGCAG), as previously described [37]. 

For further identification of the *Alternaria* spp. isolates, two additional PCR reactions were used; EF1 PCR was conducted as described above for *Fusarium* identification and RNA polymerase II (RPB2) amplification, using the RPB2-5F2 (5′-GGGGWGAYCAGAAGAAGGC) and fRPB2-7cR (5′-CCCATRGCTTGTYYRCCCAT) primers, according to Woudenberg et al. [38]. A summary of the PCR primers that were used in this study is described in Appendix A. PCR products were separated in agarose gel (0.5% TAE, 1.2% agarose (Lonza, Basel, Switzerland]) at a voltage of 80 V for 60 min [39], and documented in an ENDURE GDS apparatus (Labnet, Edison, NJ, USA). Amplification products were subsequently sequenced using standard-Seq (Macrogen, Amsterdam, The Netherlands), processed using MEGA × 10.1.7 software, and compared to existing sequences in the NCBI database using BLAST (https://blast.ncbi.nlm.nih.gov/Blast.cgi). DNA sequences of the major pathogens found in this study have been deposited in GenBank; for accession numbers, see Table 2. 

### 4.3. Koch Postulate Assays

The fungal species selected for Koch postulate assays were based on several criteria. *A. alternata*, *B. cinerea, F. solani*, and *F. oxysporum* were chosen, since they were the most common fungi isolated from affected plant tissues in Israel, and had previously been reported in the literature as major MC pathogens [1]. Other fungi, such as *Aspergillus* spp., *Cladosporium* spp., *Penicillium* spp., and *Trichothecium roseum* were selected mainly due to health hazards, since all of these fungi possess the potential to produce hazardous mycotoxins that are harmful to humans [1,33]. 

### 4.4. Production of Plant Material

*Cannabis sativa* cultivar BB 734 (derived from the shoots of third generation mother plants) was grown in the Agriculture Research Organization (ARO), Volcani Center research facility (authorized by the Israeli Medical Cannabis Agency (IMCA), Ministry of Health, State of Israel) for use in Koch postulate assays. Uncharacterized cannabis seedlings were kindly provided by Dr. Moshe Flaishman of the ARO. Growth conditions of plant material (seedlings, leaves, and mature inflorescences) were essentially as described [3]. Shoots were rooted under continuous 24 h light photo-periodic conditions of 880 lux for 1 week in a closed plastic planting container (80 × 40 × 50 cm) in a humid environment, and an additional week without the top cover. The rooted shoots were replanted in 0.2 L pots and transferred for vegetative propagation under photoperiodic conditions of 18 h light (3000 lux) and 6 h dark for 2 months. Plants were retransferred into 0.5 to 2 L pots and placed in a flowering induction chamber (4 × 3 m) for 80– 90 days. The flowering chamber contained six 600 W, high-pressure sodium lamps (SunMaster, Twinsburg, OH, USA) with dual red and blue spectrum light, under photo-periodic conditions of 11 h light (50,000 lux) and 13 h dark, until flowers were produced.

### 4.5. Koch Postulate Assays on Detached Plant Tissues

To complete Koch postulate assays, the fungi were cultured on PDAC 1.5% plates. After approximately one week, when the mycelium had completely covered the plate, 1 cm diameter sections were removed from the leading edge of the cultures and inoculated onto healthy MC tissue, i.e., fungi isolated from naturally infected MC leaves were inoculated onto healthy leaves and inflorescences. Inoculated plant tissues were placed on top of a sterile plastic net in 90 mm sterile Petri plates containing sterile paper towels soaked in 1 mL sterile distilled H_2_O, sealed with Parafilm, and kept at room temperature for 3–5 days until disease symptoms developed [9]. Plates were examined for fungal growth and disease development, and the casual agents of disease were re-isolated from diseased tissue. In these experiments, five biological repeats were conducted for each isolate. The two most common foliar pathogenic fungi (*A. alternata* and *B. cinerea*) were further tested for disease development on intact plant tissues.

### 4.6. Koch Postulate Assays on Intact Plants

The inoculation of intact plants was conducted by spraying a spore suspension of 10^6^ spores/mL for each tested isolate. Fifteen mL of sterile distilled water was added to PDAC 1.5% plates containing the tested fungus, and spores were then gently removed using a sterile rod. The liquid was then filtered through four layers of sterile gauze and collected in a 50 mL Falcon tube. Falcon tubes were centrifuged in a MEGAFUGE 16R (Heraeus, Hanau, Germany) at 900 RPM and 4 °C for 10 min. The supernatant was drained, and the pellet was suspended in 20 mL of sterile distilled H_2_O, centrifuged, and dried again. The final pellet was suspended in 15 mL sterile distilled H_2_O, the spore concentration was enumerated using a hemocytometer and a LEICA DM500 light microscope (Leica Microsystems, Wetzlar, Germany), and adjusted to 10^6^ spores/mL. Spore suspensions were sprayed on healthy MC plants at the beginning of inflorescence development (inflorescence color transparent white) until run-off, and the plants were then covered with a moist plastic bag for 3–5 days. Plants were then examined for fungal growth and disease development. 

### 4.7. Koch Postulate Assays on Seedlings for Soilborne Pathogens 

To conduct Koch postulates on MC seedlings, a spore suspension was prepared as described above from *F. oxysporum* and *F. solani*. Two-week-old MC seedlings (15 per isolate) were sampled, and roots were washed using sterile distilled H_2_O and soaked in a spore suspension for 7 min. Seedlings were then planted separately in 0.36 L pots, grown for 30 days in photoperiodic conditions of 18 h light (3000 lux) and 6 h dark, and wilt symptoms were evaluated. 

In all Koch postulate assays (detached plant tissues, intact plants, and seedlings), experiments were conducted at least twice for each tested pathogen, achieving similar results. In all tests, the fungi causing disease symptoms were re-isolated and verified by morphological characteristics, and in certain cases by molecular methods. 

## Figures and Tables

**Figure 1 plants-09-00882-f001:**
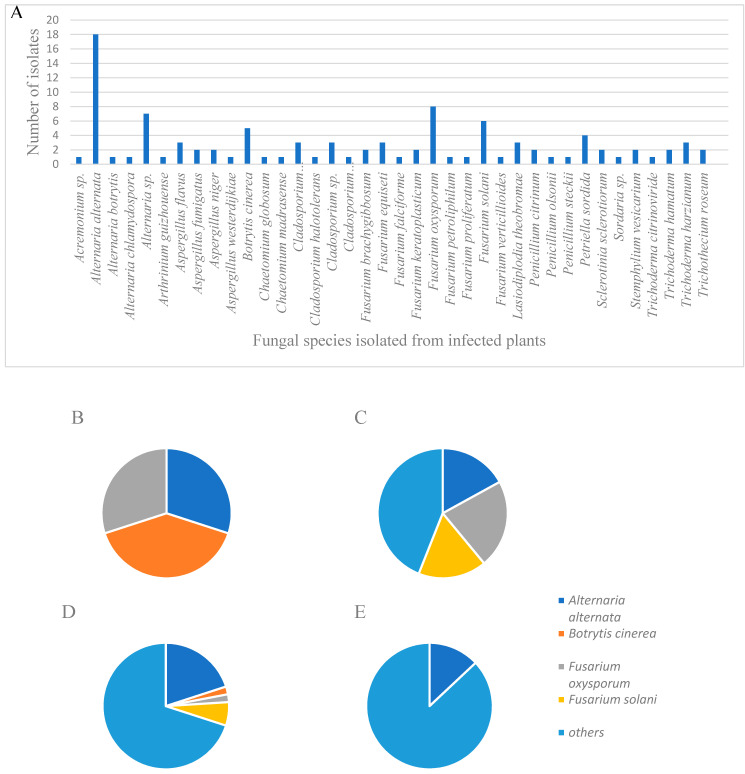
(**A**) Fungal species that were isolated from infected plant material from commercial farms throughout Israel. The four most frequent fungi that were isolated included *Alternaria alternata*, *Botrytis cinerea, Fusarium oxysporum*, and *F. solani.* (**B**–**E**) Pie charts of isolated fungi, with emphasis on the four most common fungi that were isolated (*Alternaria alternata, Botrytis cinerea, Fusarium oxysporum, F. solani* and others) during different seasons: (**B**) spring (March–May), (**C**) summer (June–August), (**D**) autumn (September–November) and (**E**) winter (December–February).

**Figure 2 plants-09-00882-f002:**
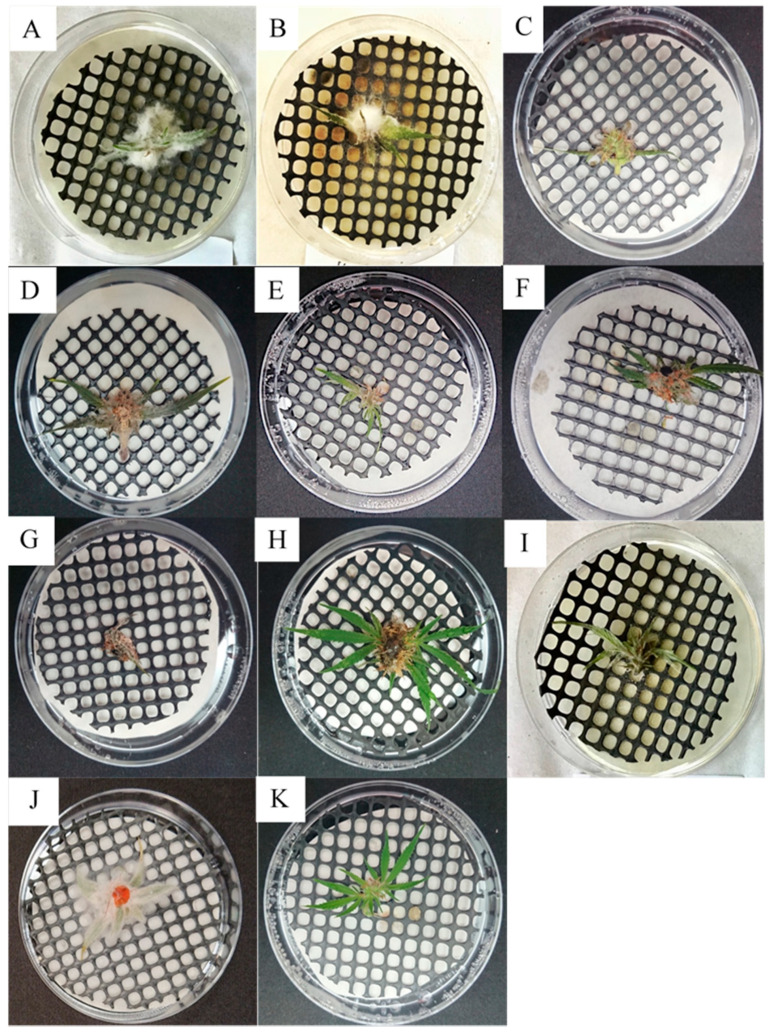
Koch postulate assays of detached MC inflorescence inoculated with (**A**) *Alternaria alternata* (isolate 22A); (**B**) *Botrytis cinerea* (isolate 63A); (**C**) *Aspergillus flavus* (isolate 49A); (**D**) *Aspergillus westerdijkiae* (isolate 71B); (**E**) *Cladosporium tenuissimum* (isolate 119B); (**F**) *Cladosporium* spp. (isolate 32A); (**G**) *Penicillium citrinum* (isolate 25A); (**H**) *Penicillium steckii* (isolate 33B); (**I**) *Trichothecium roseum* (isolate 16A); (**J**) *Fusarium equiseti* (isolate 10A); and (**K**) the control, inoculated with a sterile PDAC plug.

**Figure 3 plants-09-00882-f003:**
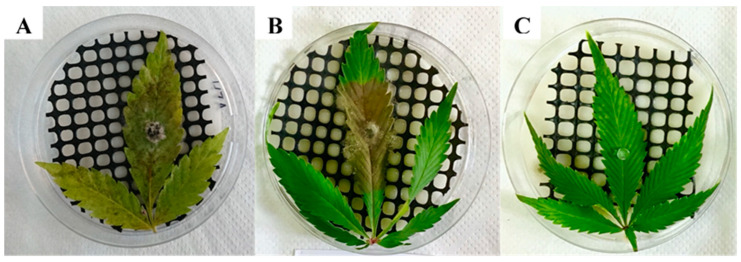
Koch postulate assays on detached MC leaves, inoculated using a PDAC plug, of (**A**) *Alternaria alternata* (isolate 22A); (**B**) *Botrytis cinerea* (isolate 63A); and (**C**) control leaves, inoculated with a sterile PDAC plug.

**Figure 4 plants-09-00882-f004:**
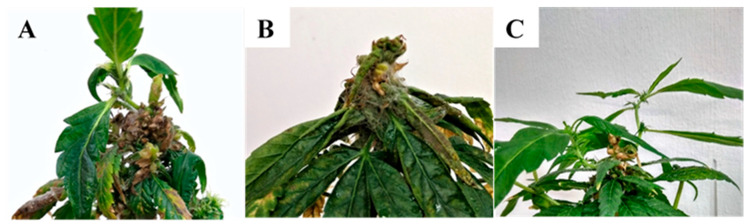
Koch postulate assays of intact MC plants sprayed with a 10^6^ spores/mL spore suspension of (**A**) *Alternaria alternata* (isolate 22A), (**B**) *Botrytis cinerea* (isolate 63A), and (**C**) a control plant sprayed with sterile distilled H_2_O.

**Figure 5 plants-09-00882-f005:**
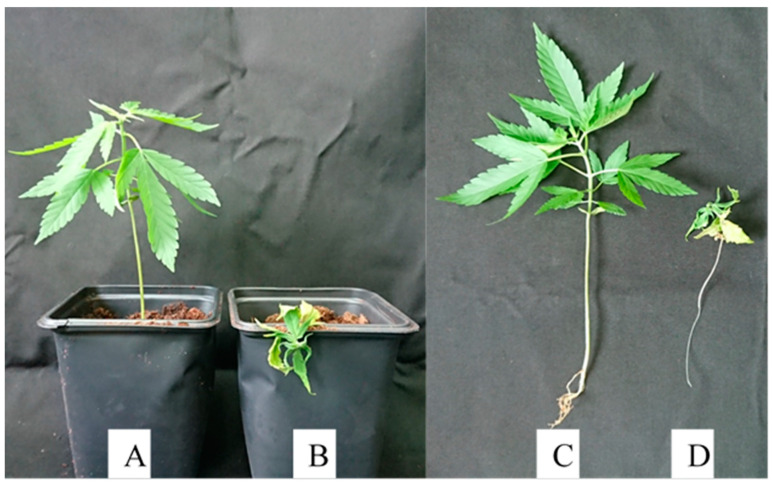
Inoculation of an MC seedling with (**A**) saline solution (control) compared to that inoculated with (**B**) *Fusarium oxysporum* (isolate 64B). Images of intact plants, including roots of (**C**) a plant inoculated with saline solution (control) compared to that (**D**) inoculated with *Fusarium oxysporum* (isolate 64B).

**Table 1 plants-09-00882-t001:** Summary of Koch postulate assays for each fungal species.

Koch Postulate Assay ^a^			
Isolate	Species	Intact Plant	Detached Tissue
22A	*Alternaria alternata*	+	+
49A	*Aspergillus flavus*	ND ^b^	+
106A	*Aspergillus fumigatus*	ND	−
53A	*Aspergillus niger*	ND	−
71B	*Aspergillus westerdijkiae*	ND	+
63A	*Botrytis cinerea*	+	+
36B	*Cladosporium cladosporioides*	ND	+
35A	*Cladosporium halotolerans*	ND	−
32A	*Cladosporium* spp.	ND	+
42A	*Cladosporium sphaerospermum*	ND	−
119B	*Cladosporium tenuissimum*	ND	+
10A	*Fusarium equiseti*	ND	+
64B	*Fusarium oxysporum*	+	ND
99A	*Fusarium proliferatum*	+	ND
109A-3	*Fusarium solani*	+	ND
25A	*Penicillium citrinum*	ND	+
110A	*Penicillium olsonii*	ND	−
33B	*Penicillium steckii*	ND	+
16A	*Trichothecium roseum*	ND	+

^a^ “+” signifies a positive response (appearance of symptoms), while “–“ signifies a negative response or lack of symptom appearance. ^b^ ND: not determined.

**Table 2 plants-09-00882-t002:** Primers used for fungal identification of the most frequent pathogens of medical cannabis (MC) (Figure 1) in this study, and their corresponding accession numbers deposited in GenBank.

		Primers Used for Fungal Identification with Corresponding Accession Numbers			
Isolate Designation	Fungal Species	ITS 4-5	EF1	RPB2	IGS
22A	*Alternaria alternata*	MT180724	MT186261	MT186265	– ^a^
62A	*Fusarium oxysporum*	MT180726	MT186263	–	MT199205
63A	*Botrytis cinerea*	MT180725	–	–	–
64B	*Fusarium oxysporum*	MT180727	MT186262	–	MT199206
109A-3	*Fusarium solani*	MT229135	MT254546	–	–

^a^ Primers not used for pathogen identification.

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
