# Peer review of "Fungal Pathogens Affecting the Production and Quality of Medical Cannabis in Israel"

_plants, 2020, doi:10.3390/plants9070882_

Round 1
Reviewer 1 Report
The paper is well written and I think it gives important information about fungal pathogens of medical cannabis with potential negative effects on consumers.
I only have some recommendations for authors.
The main problems for me were tables. I did not found table 1 in the paper but only supplementary table 1, moreover table 3 is cited before table 1 and 2. Please check the order of the tables.
I also found a table at line 285 without any title nor number. At line 305 there is a table title without the table. Maybe there is some confusing about tables. Please check them carefully.
I found in the M&M section many references to table 2 that is a result table. I think that all these references has to be deleted from the M&M section (for example at lines 328 and 332).
Regarding Figure 1 I think it could better to refer to % of isolates and not to number of isolates.
Line 98: title of the paragraph was repeated at the beginning of the paragraph. Please delete this repetition.
Line 99: A. niger and not Aspergillus niger
Line 98: as already reported, Table 3 results before table 1 and 2. Please order table correctly.
Line 114: delete the empty line 115
Line 118: the font used is bigger than the rest of the tet. Please correct it
Line 139: Fusarium oxysporum, F. proliferatum and F. solani
Lines 270-271: please explain briefly the methodology for DNA extraction.
Line 283: as PREVIOUSLY described [and please remove (.] Moreover I think that all methodologies have to be briefly described.
Line 288: add . after [17].
Line 315: please describe shortly the methodology.
Lines 330-334: at which phenological stage were sprayed plants? Please add this information.
Author Response
The main problems for me were tables. I did not found table 1 in the paper but only supplementary table 1, moreover table 3 is cited before table 1 and 2. Please check the order of the tables.
Tables were renumbered and citation order arranged as suggested.
I also found a table at line 285 without any title nor number. At line 305 there is a table title without the table. Maybe there is some confusing about tables. Please check them carefully.
Corrected.
I found in the M&M section many references to table 2 that is a result table. I think that all these references has to be deleted from the M&M section (for example at lines 328 and 332).
Citations of Table 1 (previously Table 2 before rearrangement) were removed from the M&M section.
Regarding Figure 1 I think it could better to refer to % of isolates and not to number of isolates.
The results in numbers and not percentages are better represented since percentages generate a less accurate picture especially concerning certain pathogens that were found in low quantities but have a significant adverse effect regarding human health.
Line 98: title of the paragraph was repeated at the beginning of the paragraph. Please delete this repetition.
The title was deleted from the beginning of the paragraph.
Line 99: A. niger and not Aspergillus niger
Replaced.
Line 98: as already reported, Table 3 results before table 1 and 2. Please order table correctly.
Order of tables corrected.
Line 114: delete the empty line 115
Deleted.
Line 118: the font used is bigger than the rest of the tet. Please correct it
Corrected.
Line 139: Fusarium oxysporum, F. proliferatum and F. solani
Corrected.
Lines 270-271: please explain briefly the methodology for DNA extraction.
DNA extraction methodology was added.
Line 283: as PREVIOUSLY described [and please remove (.]
Corrected.
Moreover I think that all methodologies have to be briefly described.
Methodologies were more thoroughly described.
Line 288: add . after [17].
Added.
Line 315: please describe shortly the methodology.
Methodology added.
Lines 330-334: at which phenological stage were sprayed plants? Please add this information.
Phenological stage of sprayed plants added.

Reviewer 2 Report
The paper “Fungal Pathogens Affecting the Production and Quality of Medical Cannabis in Israel” is an interesting manuscript and it fits with the aim of the journal. In my opinion it can be considered for the pubblication after minor revision. Introduction provides a correct background with adeguate references; Section are well organized and detailed. Just few comments. The ends 5 'and 3' n the sequence of primers used should be indicated;
check the name of subsp should be writter not with capital letter;
Line 98: check the sentences: "Koch postulate assays in detached plant tissues The fungi that were tested are specified in Table 3"
Put together table 3 and its caption and foodnote
Line 114-116 check, there is an empty line
Line 139:add comma between Fusarium oxysporum F. proliferatum
g/l=g/L
Author Response
The ends 5 'and 3' n the sequence of primers used should be indicated;
Added.
check the name of subsp should be written not with capital letter;
Corrected.
Line 98: check the sentences: "Koch postulate assays in detached plant tissues The fungi that were tested are specified in Table 3"
Corrected.
Put together table 3 and its caption and footnote
Corrected.
Line 114-116 check, there is an empty line
Corrected.
Line 139:add comma between Fusarium oxysporum F. proliferatum
Added.
g/l=g/L
Corrected.

Reviewer 3 Report
S. Jerushalmi and colleagues present an article on the isolation of fungi from infected medical Cannabis plants in commercial farms in Israel. In general the manuscript is comprehensibly written and may provide interesting information for people working in this field. Authors should however work more on the presentation of the article by substantially improving the quality of provided figures.
Images showing fungal growth on petri dishes and perhaps some general morphological features of at least some of the isolated fungi can be added.
Figure 2 may be presented in color and combined with Figure 1.
Phenotypes shown in Figure 3 are difficult to distinguish in this background.
Figure 6 should be replaced.
Authors should provide, or refer to, more information related to the sample areas, like for example specific climatic, soil and cultivation parameters. In this article, the sampling areas are very specific and in this context, authors should be careful when referring to “…the most frequent pathogens of MC in Israel…”
Authors may want to consider avoiding the repetition of the words “Koch’s postulates”, especially in the chapter titles and legends, and instead describe the results shown.
Authors state that they have compared the DNA sequences of the isolated strains to existing sequences in NCBI database. They have however decided to deposit only a few of those in GenBank. Were these the only ones with differences in their sequence? Authors may want to consider commenting on these differences.
Line 98: The title of the chapter appears twice in the text.
Line 305: Table 3 legend appears misplaced.
Author Response
Images showing fungal growth on petri dishes and perhaps some general morphological features of at least some of the isolated fungi can be added.
As the main pathogens we found in this study are common and well known fungi we believe that adding such a figure is not crucial and informative in the context of this research.
Figure 2 may be presented in color and combined with Figure 1.
Figure 2 was changed to color and combined with Figure 1.
Phenotypes shown in Figure 3 are difficult to distinguish in this background.
Unfortunately, pictures with the same background are the only ones we possess, but we will implement this comment in future experiments.
Figure 6 should be replaced.
Figure 6 was replaced, but is now numbered Figure 5…
Authors should provide, or refer to, more information related to the sample areas, like for example specific climatic, soil and cultivation parameters.
Strict confidentially terms have been agreed upon in collaboration with the MC farms we worked with. Therefore, we cannot disclose these details without infringing terms of this agreement.
In this article, the sampling areas are very specific and in this context, authors should be careful when referring to “…the most frequent pathogens of MC in Israel…”
Changed to "… most frequent pathogens of MC found in this study…"
Authors may want to consider avoiding the repetition of the words “Koch’s postulates”, especially in the chapter titles and legends, and instead describe the results shown.
The use of the term “Koch’s postulates” was reduced.
Authors state that they have compared the DNA sequences of the isolated strains to existing sequences in NCBI database. They have however decided to deposit only a few of those in GenBank. Were these the only ones with differences in their sequence? Authors may want to consider commenting on these differences.
We uploaded only sequences belonging to the actual representative isolates used in the Koch's postulate assays. Others were identical therefore were not submitted to GenBank.
Line 98: The title of the chapter appears twice in the text.
Title deleted from text.
Line 305: Table 3 legend appears misplaced.
Corrected.

Round 2
Reviewer 3 Report
The revised version of the article by Jerushalmi et al has improved considerably, although, in my opinion, the overall presentation would benefit from further optimization of certain figures.
Author Response
- The statement has been inserted in the Acknowledgement section.
- The amount of chloramphenicol is correct
